# Provably Efficient Exploration for Reinforcement Learning Using Unsupervised Learning [*]

**Fei Feng**
University of California, Los Angeles
fei.feng@math.ucla.edu

**Ruosong Wang**
Carnegie Mellon University
ruosongw@andrew.cmu.edu

**Wotao Yin**
University of California, Los Angeles
wotaoyin@math.ucla.edu

**Simon S. Du**
University of Washington
ssdu@cs.washington.edu

**Lin F. Yang**
University of California, Los Angeles
linyang@ee.ucla.edu

## Abstract

Motivated by the prevailing paradigm of using unsupervised learning for efficient exploration in reinforcement learning (RL) problems (Tang et al., 2017; Bellemare et al., 2016), we investigate when this paradigm is provably efficient. We study episodic Markov decision processes with rich observations generated from a small number of latent states. We present a general algorithmic framework that is built upon two components: an unsupervised learning algorithm and a no-regret tabular RL algorithm. Theoretically, we prove that as long as the unsupervised learning algorithm enjoys a polynomial sample complexity guarantee, we can find a near-optimal policy with sample complexity polynomial in the number of latent states, which is significantly smaller than the number of observations. Empirically, we instantiate our framework on a class of hard exploration problems to demonstrate the practicality of our theory.

## 1 Introduction

Reinforcement learning (RL) is the framework of learning to control an unknown system through trial and error. It takes as inputs the observations of the environment and outputs a policy, i.e., a mapping from observations to actions, to maximize the cumulative rewards. To learn a near-optimal policy, it is critical to sufficiently explore the environment and identify all the opportunities for high rewards. However, modern RL applications often need to deal with huge observation spaces such as those consist of images or texts, which makes it challenging or impossible (if there are infinitely many observations) to fully explore the environment in a direct way. In some work, function approximation scheme is adopted such that essential quantities for policy improvement, e.g. state-action values, can be generalized from limited observed data to the whole observation space. However, the use of function approximation alone does not resolve the exploration problem (Du et al., 2020a).

To tackle this issue, multiple empirically successful strategies are developed (Tang et al., 2017; Bellemare et al., 2016; Pathak et al., 2017; Azizzadenesheli et al., 2018; Lipton et al., 2018; Fortunato et al., 2018; Osband et al., 2016). Particularly, in Tang et al. (2017) and Bellemare et al. (2016), the

---

[*]Correspondence to: Simon S. Du <ssdu@cs.washington.edu>, Lin F. Yang <linyang@ee.ucla.edu>

authors use state abstraction technique to reduce the problem size. They construct a mapping from observations to a small number of hidden states and devise exploration on top of the latent state space rather than the original observation space.

To construct such a state abstraction mapping, practitioners often use *unsupervised learning*. The procedure has the following steps: collect a batch of observation data, apply unsupervised learning to build a mapping, use the mapping to guide exploration and collect more data, and repeat. Empirical study evidences the effectiveness of such an approach at addressing hard exploration problems (e.g., the infamous Montezuma's Revenge). However, it has not been theoretically justified. In this paper, we aim to answer this question:

*Is exploration driven by unsupervised learning in general **provably efficient**?*

The generality includes the choice of unsupervised learning algorithms, reinforcement learning algorithms, and the condition of the problem structure.

We first review some existing theoretical results on provably efficient exploration. More discussion about related work is deferred to appendix. For an RL problem with finitely many states, there are many algorithms with a tabular implementation that learn to control efficiently. These algorithms can learn a near-optimal policy using a number of samples polynomially depending on the size of the state space. However, if we directly apply these algorithms to rich observations cases by treating each observation as a state, the sample complexities are polynomial in the cardinality of the observation space. Such a dependency is unavoidable without additional structural assumptions (Jaksch et al., 2010). If structural conditions are considered, for example, observations are generated from a small number of latent states (Krishnamurthy et al., 2016; Jiang et al., 2017; Dann et al., 2018; Du et al., 2019a), then the sample complexity only scales polynomially with the number of hidden states. Unfortunately, the correctness of these algorithms often requires strict assumptions (e.g., deterministic transitions, reachability) that may not be satisfied in many real applications.

**Our Contributions** In this paper we study RL problems with rich observations generated from a small number of latent states for which an unsupervised learning subroutine is used to guide exploration. We summarize our contributions below.

- We propose a new algorithmic framework for the Block Markov Decision Process (BMDP) model (Du et al., 2019a). We combine an unsupervised learning oracle and a tabular RL algorithm in an organic way to find a near-optimal policy for a BMDP. The unsupervised learning oracle is an abstraction of methods used in Tang et al. (2017); Bellemare et al. (2016) and widely used statistical generative models. Notably, our framework can take almost *any* unsupervised learning algorithms and tabular RL algorithms as subroutines.
- Theoretically, we prove that as long as the unsupervised learning oracle and the tabular RL algorithm each has a polynomial sample complexity guarantee, our framework finds a near-optimal policy with sample complexity polynomial in the number of latent states, which is significantly smaller than the number of possible observations (cf. Theorem 1). To our knowledge, this is the *first* provably efficient method for RL problems with huge observation spaces that uses unsupervised learning for exploration. Furthermore, our result does not require additional assumptions on transition dynamics as used in Du et al. (2019a). Our result theoretically sheds light on the success of the empirical paradigms used in Tang et al. (2017); Bellemare et al. (2016).
- We instantiate our framework with particular unsupervised learning algorithms and tabular RL algorithms on hard exploration environments with rich observations studied in Du et al. (2019a), and compare with other methods tested in Du et al. (2019a). Our experiments demonstrate our method can significantly outperform existing methods on these environments.

**Main Challenge and Our Technique** We assume there is an unsupervised learning oracle (see formal definition in Section 4) which can be applied to learn decoding functions and the accuracy of learning increases as more training data are fed. The unsupervised learning algorithm can only guarantee good performance with respect to the input distribution that generates the training data. Unlike standard unsupervised learning where the input distribution is fixed, in our problem, the input distribution depends on our policy. On the other hand, the quality of a policy depends on whether the unsupervised learning oracle has (approximately) decoded the latent states. This interdependency is the main challenge we need to tackle in our algorithm design and analysis.

Here we briefly explain our framework. Let $\mathcal{M}$ be the MDP with rich observations. We form an auxiliary MDP $\mathcal{M}'$ whose state space is the latent state space of $\mathcal{M}$. Our idea is to simulate the process of running a no-regret tabular RL algorithm $\mathscr{A}$ directly on $\mathcal{M}'$. For each episode, $\mathscr{A}$ proposes a policy $\pi$ for $\mathcal{M}'$ and expects a trajectory of running $\pi$ on $\mathcal{M}'$ for updating and then proceeds. To obtain such a trajectory, we design a policy $\phi$ for $\mathcal{M}$ as a composite of $\pi$ and some initial decoding functions. We run $\phi$ on $\mathcal{M}$ to collect observation trajectories. Although the decoding functions may be inaccurate initially, they can still help us collect observation samples for later refinement. After collecting sufficient observations, we apply the unsupervised learning oracle to retrain decoding functions and then update $\phi$ as a composite of $\pi$ and the newly-learned functions and repeat running $\phi$ on $\mathcal{M}$. After a number of iterations (proportional to the size of the latent state space), with the accumulation of training data, decoding functions are trained to be fairly accurate on recovering latent states, especially those $\pi$ has large probabilities to visit. This implies that running the latest $\phi$ on $\mathcal{M}$ is almost equivalent to running $\pi$ on $\mathcal{M}'$. Therefore, we can obtain a state-action trajectory with high accuracy as the algorithm $\mathscr{A}$ requires. Since $\mathscr{A}$ is guaranteed to output a near-optimal policy after a polynomial (in the size of the true state-space) number of episodes, our algorithm uses polynomial number of samples as well.

## 2   Related Work

In this section, we review related provably efficient RL algorithms. We remark that we focus on environments that require explicit exploration. With certain assumptions of the environment, e.g., the existence of a good exploration policy or the distribution over the initial state is sufficiently diverse, one does not need to explicitly explore (Munos, 2005; Antos et al., 2008; Geist et al., 2019; Kakade and Langford, 2002; Bagnell et al., 2004; Scherrer and Geist, 2014; Agarwal et al., 2020; Fan et al., 2020; Chen and Jiang, 2019). Without these assumptions, the problem can require an exponential number of samples, especially for policy-based methods (Du et al., 2020a).

Exploration is needed even in the most basic tabular setting. There is a substantial body of work on provably efficient tabular RL (Agrawal and Jia, 2017; Jaksch et al., 2010; Kakade et al., 2018; Azar et al., 2017; Kearns and Singh, 2002; Dann et al., 2017; Strehl et al., 2006; Jin et al., 2018; Simchowitz and Jamieson, 2019; Zanette and Brunskill, 2019). A common strategy is to use UCB bonus to encourage exploration in less-visited states and actions. One can also study RL in metric spaces (Pazis and Parr, 2013; Song and Sun, 2019; Ni et al., 2019). However, in general, this type of algorithms has an exponential dependence on the state dimension.

To deal with huge observation spaces, one might use function approximation. Wen and Van Roy (2013) proposed an algorithm, optimistic constraint propagation (OCP), which enjoys polynomial sample complexity bounds for a family of $Q$-function classes, including the linear function class as a special case. But their algorithm can only handle deterministic systems, i.e., both transition dynamics and rewards are deterministic. The setting is recently generalized by Du et al. (2019b) to environments with low variance and by Du et al. (2020b) to the agnostic setting. Li et al. (2011) proposed a Q-learning algorithm which requires the Know-What-It-Knows oracle. But it is in general unknown how to implement such an oracle.

Our work is closely related to a sequence of works which assumes the transition has certain low-rank structure (Krishnamurthy et al., 2016; Jiang et al., 2017; Dann et al., 2018; Sun et al., 2019; Du et al., 2019a; Jin et al., 2020; Yang and Wang, 2019). The most related paper is Du et al. (2019a) which also builds a state abstraction map. Their sample complexity depends on two quantities of the transition probability of the hidden states: *identifiability* and *reachability*, which may not be satisfied in many scenarios. Identifiability assumption requires that the $L_1$ distance between the posterior distributions (of previous level's hidden state, action pair) given any two different hidden states is strictly larger than some constant (Assumption 3.2 in Du et al. (2019a)). This is an inherent necessary assumption for the method in Du et al. (2019a) as they need to use the posterior distribution to distinguish hidden states. Reachability assumption requires that there exists a constant such that for every hidden state, there exists a policy that reaches the hidden state with probability larger than this constant (Definition 2.1 in Du et al. (2019a)). Conceptually, this assumption is not needed for finding a near-optimal policy because if one hidden state has negligible reaching probability, one can just ignore it. Nevertheless, in Du et al. (2019a), the reachability assumption is also tied with building the abstraction map. Therefore, it may not be removable if one uses the strategy in Du et al.

(2019a). In this paper, we show that given an unsupervised learning oracle, one does not need the identifiability and reachability assumptions for efficient exploration.

## 3 Preliminaries

**Notations** Given a set $\mathcal{A}$, we denote by $|\mathcal{A}|$ the cardinality of $\mathcal{A}$, $\mathcal{P}(\mathcal{A})$ the set of all probability distributions over $\mathcal{A}$, and $\mathrm{Unif}(\mathcal{A})$ the uniform distribution over $\mathcal{A}$. We use $[h]$ for the set $\{1, 2, \ldots, h\}$ and $f_{[h]}$ for the set of functions $\{f_1, f_2, \ldots, f_h\}$. Given two functions $f : \mathcal{X} \to \mathcal{Y}$ and $g : \mathcal{Y} \to \mathcal{Z}$, their composite is denoted as $g \circ f : \mathcal{X} \to \mathcal{Z}$.

**Block Markov Decision Process** We consider a Block Markov Decision Process (BMDP), which is first formally introduced in Du et al. (2019a). A BMDP is described by a tuple $\mathcal{M} := (\mathcal{S}, \mathcal{A}, \mathcal{X}, \mathcal{P}, r, f_{[H+1]}, H)$. $\mathcal{S}$ is a finite unobservable *latent state space*, $\mathcal{A}$ is a finite action space, and $\mathcal{X}$ is a possibly infinite observable context space. $\mathcal{X}$ can be partitioned into $|\mathcal{S}|$ disjoint blocks $\{\mathcal{X}_s\}_{s \in \mathcal{S}}$, where each block $\mathcal{X}_s$ corresponds to a unique state $s$. $\mathcal{P}$ is the collection of the *state-transition probability* $p_{[H]}(s'|s, a)$ and the *context-emission distribution* $q(x|s)$ for all $s, s' \in \mathcal{S}, a \in \mathcal{A}, x \in \mathcal{X}$. $r : [H] \times \mathcal{S} \times \mathcal{A} \to [0, 1]$ is the reward function. $f_{[H+1]}$ is the set of decoding functions, where $f_h$ maps every observation at level $h$ to its true latent state. Finally, $H$ is the length of horizon. When $\mathcal{X} = \mathcal{S}$, this is the usual MDP setting.

For each episode, the agent starts at level 1 with the initial state $s_1$ and takes $H$ steps to the final level $H + 1$. We denote by $\mathcal{S}_h$ and $\mathcal{X}_h$ the set of possible states and observations at level $h \in [H + 1]$, respectively. At each level $h \in [H + 1]$, the agent has no access to the true latent state $s_h \in \mathcal{S}_h$ but an observation $x_h \sim q(\cdot|s_h)$. An action $a_h$ is then selected following some policy $\phi : [H] \times \mathcal{X} \to \mathcal{P}(\mathcal{A})$. As a result, the environment evolves into a new state $s_{h+1} \sim p_h(\cdot|s_h, a_h)$ and the agent receives an instant reward $r(h, s_h, a_h)$. A trajectory has such a form: $\{s_1, x_1, a_1, \ldots, s_H, x_H, a_H, s_{H+1}, x_{H+1}\}$, where all state components are unknown.

**Policy** Given a BMDP $\mathcal{M} := (\mathcal{S}, \mathcal{A}, \mathcal{X}, \mathcal{P}, r, f_{[H+1]}, H)$, there is a corresponding MDP $\mathcal{M}' := (\mathcal{S}, \mathcal{A}, \mathcal{P}, r, H)$, which we refer to as *the underlying MDP* in the later context. A policy on $\mathcal{M}$ has a form $\phi : [H] \times \mathcal{X} \to \mathcal{P}(\mathcal{A})$ and a policy on $\mathcal{M}'$ has a form $\pi : [H] \times \mathcal{S} \to \mathcal{P}(\mathcal{A})$. Given a policy $\pi$ on $\mathcal{M}'$ and a set of functions $\hat{f}_{[H+1]}$ where $\hat{f}_h : \mathcal{X}_h \to \mathcal{S}_h, \forall h \in [H + 1]$, we can induce a policy on $\mathcal{M}$ as $\pi \circ \hat{f}_{[H+1]} =: \phi$ such that $\phi(h, x_h) = \pi(h, \hat{f}_h(x_h)), \forall x_h \in \mathcal{X}_h, h \in [H]$. If $\hat{f}_{[H+1]} = f_{[H+1]}$, then $\pi$ and $\phi$ are equivalent in the sense that they induce the same probability measure over the state-action trajectory space.

Given an MDP, the value of a policy $\pi$ (starting from $s_1$) is defined as $V_1^\pi = \mathbb{E}^\pi \left[ \sum_{h=1}^{H} r(h, s_h, a_h) \middle| s_1 \right]$, A policy that has the maximal value is an *optimal policy* and the *optimal value* is denoted by $V_1^*$, i.e., $V_1^* = \max_\pi V_1^\pi$. Given $\varepsilon > 0$, we say $\pi$ is $\varepsilon$-*optimal* if $V_1^* - V_1^\pi \leq \varepsilon$. Similarly, given a BMDP, we define the value of a policy $\phi$ (starting from $s_1$) as: $V_1^\phi = \mathbb{E}^\phi \left[ \sum_{h=1}^{H} r(h, s_h, a_h) \middle| s_1 \right]$, The notion of optimallity and $\varepsilon$-optimality are similar to MDP.

## 4 A Unified Framework for Unsupervised Reinforcement Learning

### 4.1 Unsupervised Learning Oracle and No-regret Tabular RL Algorithm

In this paper, we consider RL on a BMDP. The goal is to find a near-optimal policy with sample complexity polynomial to the cardinality of the latent state space. We assume no knowledge of $\mathcal{P}$, $r$, and $f_{[H+1]}$, but the access to an unsupervised learning oracle $\mathcal{ULO}$ and an $(\varepsilon, \delta)$-correct episodic no-regret algorithm. We give the definitions below.

**Definition 1** (Unsupervised Learning Oracle $\mathcal{ULO}$)**.** *There exists a function $g(n, \delta)$ such that for any fixed $\delta > 0$, $\lim_{n \to \infty} g(n, \delta) = 0$. Given a distribution $\mu$ over $\mathcal{S}$, and $n$ samples from $\sum_{s \in \mathcal{S}} q(\cdot|s)\mu(s)$, with probability at least $1 - \delta$, we can find a function $\hat{f} : \mathcal{X} \to \mathcal{S}$ such that*

$$\mathbb{P}_{s \sim \mu, x \sim q(\cdot|s)} \big( \hat{f}(x) = \alpha(s) \big) \geq 1 - g(n, \delta)$$

*for some unknown permutation $\alpha : \mathcal{S} \to \mathcal{S}$.*

---

**Algorithm 1** A Unified Framework for Unsupervised RL

---

1: **Input:** BMDP $\mathcal{M}$; $\mathcal{ULO}$; $(\varepsilon, \delta)$-correct episodic no-regret algorithm $\mathscr{A}$; batch size $B > 0$; $\varepsilon \in (0,1)$; $\delta \in (0,1)$; $N := \lceil \log(2/\delta)/2 \rceil$; $L := \lceil 9H^2/(2\varepsilon^2) \log(2N/\delta) \rceil$.

2: **for** $n = 1$ **to** $N$ **do**

3:     Clear the memory of $\mathscr{A}$ and restart;

4:     **for** episode $k = 1$ **to** $K$ **do**

5:         Obtain $\pi^k$ from $\mathscr{A}$;

6:         Obtain a trajectory: $\tau^k, f^k_{[H+1]} \leftarrow \text{TSR}(\mathcal{ULO}, \pi^k, B)$;

7:         Update the algorithm: $\mathscr{A} \leftarrow \tau^k$;

8:     **end for**

9:     Obtain $\pi^{K+1}$ from $\mathscr{A}$;

10:     Finalize the decoding functions: $\tau^{K+1}, f^{K+1}_{[H+1]} \leftarrow \text{TSR}(\mathcal{ULO}, \pi^{K+1}, B)$;

11:     Construct a policy for $\mathcal{M} : \phi^n \leftarrow \pi^{K+1} \circ f^{K+1}_{[H+1]}$.

12: **end for**

13: Run each $\phi^n$ ($n \in [N]$) for $L$ episodes and get the average rewards per episode $\bar{V}_1^{\phi^n}$.

14: Output a policy $\phi \in \text{argmax}_{\phi \in \phi^{[N]}} \bar{V}_1^{\phi}$.

---

In Definition 1, suppose $f$ is the true decoding function, i.e., $\mathbb{P}_{s \sim \mu, x \sim q(\cdot|s)}\big(f(x) = s\big) = 1$. We refer to the permutation $\alpha$ as a *good permutation* between $f$ and $\hat{f}$. Given $g(n, \delta)$, we define $g^{-1}(\epsilon, \delta) := \min\{N \mid \text{for all } n > N, g(n, \delta) < \epsilon\}$. Since $\lim_{n \to \infty} g(n, \delta) = 0$, $g^{-1}(\epsilon, \delta)$ is well-defined. We assume that $g^{-1}(\epsilon, \delta)$ is a polynomial in terms of $1/\epsilon, \log(\delta^{-1})$ and possibly problem-dependent parameters.

This definition is motivated by Tang et al. (2017) in which authors use auto-encoder and SimHash (Charikar, 2002) to construct the decoding function and they use this UCB-based approach on top of the decoding function to guide exploration. It is still an open problem to obtain a sample complexity analysis for auto-encoder. Let alone the composition with SimHash. Nevertheless, in Appendix B, we give several examples of $\mathcal{ULO}$ with theoretical guarantees. Furthermore, once we have an analysis of auto-encoder and we can plug-in that into our framework effortlessly.

**Definition 2** (($\varepsilon, \delta$)-correct No-regret Algorithm). *Let $\varepsilon > 0$ and $\delta > 0$. $\mathscr{A}$ is an $(\varepsilon, \delta)$-correct no-regret algorithm if for any MDP $\mathcal{M}' := (\mathcal{S}, \mathcal{A}, \mathcal{P}, r, H)$ with the initial state $s_1$, $\mathscr{A}$*

- *runs for at most $\text{poly}(|\mathcal{S}|, |\mathcal{A}|, H, 1/\varepsilon, \log(1/\delta))$ episodes (the sample complexity of $\mathscr{A}$);*
- *proposes a policy $\pi^k$ at the beginning of episode $k$ and collects a trajectory of $\mathcal{M}'$ following $\pi^k$;*
- *outputs a policy $\pi$ at the end such that with probability at least $1 - \delta$, $\pi$ is $\varepsilon$-optimal.*

Definition 2 simply describes tabular RL algorithms that have polynomial sample complexity guarantees for episodic MDPs. Instances are vivid in literature (see Section 2).

## 4.2 A Unified Framework

With a $\mathcal{ULO}$ and an $(\varepsilon, \delta)$-correct no-regret algorithm $\mathscr{A}$, we propose a unified framework in Algorithm 1. Note that we use $\mathcal{ULO}$ as a black-box oracle for abstraction and generality. For each episode, we combine the policy $\pi$ proposed by $\mathscr{A}$ for the underlying MDP together with certain decoding functions $\hat{f}_{[H+1]}$[2] to generate a policy $\pi \circ \hat{f}_{[H+1]}$ for the BMDP. Then we collect observation samples using $\pi \circ \hat{f}_{[H+1]}$ and all previously generated policies over the BMDP. As more samples are collected, we refine the decoding functions via $\mathcal{ULO}$. Once the sample number is enough, a trajectory of $\pi \circ \hat{f}_{[H+1]}$ is as if obtained using the true decoding functions (up to some permutations). Therefore, we successfully simulate the process of running $\pi$ directly on the underlying MDP. We then proceed to the next episode with the latest decoding functions and repeat the above procedure

**Algorithm 2** Trajectory Sampling Routine TSR $(\mathcal{ULO}, \pi, B)$

---

1: **Input:** $\mathcal{ULO}$; a policy $\pi : [H] \times \mathcal{S} \to \mathcal{A}$; episode index $k$; batch size $B > 0$; $\epsilon \in (0,1)$; $\delta_1 \in (0,1)$; $J := (H+1)|\mathcal{S}| + 1$.

2: **Data:**
 • a policy set $\Pi$;
 • label standard data $\mathcal{Z} := \{\mathcal{Z}_1, \mathcal{Z}_2, \ldots \mathcal{Z}_{H+1}\}$, $\mathcal{Z}_h := \{\mathcal{D}_{h,s_1}, \mathcal{D}_{h,s_2}, \ldots\}$;
 • present decoding functions $f^0_{[H+1]}$;

3: **for** $i = 1$ **to** $J$ **do**

4:      Combine policy: $\Pi \leftarrow \Pi \cup \{\pi \circ f^{i-1}_{[H+1]}\}$;

5:      Generate $((k-1)J + i) \cdot B$ trajectories of training data $\mathcal{D}$ with $\mathrm{Unif}(\Pi)$;

6:      Generate $B$ trajectories of testing data $\mathcal{D}''$ with $\pi \circ f^{i-1}_{[H+1]}$.

7:      Train with $\mathcal{ULO}$: $\tilde{f}^i_{[H+1]} \leftarrow \mathcal{ULO}(\mathcal{D})$;

8:      Match labels: $f^i_{[H+1]} \leftarrow \mathrm{FixLabel}(\tilde{f}^i_{[H+1]}, \mathcal{Z})$;

9:      **for** $h \in [H+1]$ **do**

10:          Let $\mathcal{D}''_{h,s} := \{x \in \mathcal{D}''_h : f^i_h(x) = s, s \in \mathcal{S}_h\}$;

11:          Update label standard set: if $\mathcal{D}_{h,s} \notin \mathcal{Z}_h$ and $|\mathcal{D}''_{h,s}| \geq 3\epsilon \cdot B \log(\delta_1^{-1})$, then let $\mathcal{Z}_h \leftarrow \mathcal{Z}_h \cup \{\mathcal{D}''_{h,s}\}$

12:      **end for**

13: **end for**

14: Run $\pi \circ f^J_{[H+1]}$ to obtain a trajectory $\tau$;

15: Renew $f^0_{[H+1]} \leftarrow f^J_{[H+1]}$;

16: **Output:** $\tau, f^J_{[H+1]}$.

---

---

**Algorithm 3** FixLabel$(\tilde{f}_{[H+1]}, \mathcal{Z})$

---

1: **Input:** decoding functions $\tilde{f}_{[H+1]}$; a set of label standard data $\mathcal{Z} := \{\mathcal{Z}_1, \mathcal{Z}_2, \cdots, \mathcal{Z}_{H+1}\}$, $\mathcal{Z}_h := \{\mathcal{D}_{h,s_1}, \mathcal{D}_{h,s_2}, \ldots\}$.

2: **for** every $\mathcal{D}_{h,s}$ in $\mathcal{Z}$ **do**

3:      **if** $s \in \mathcal{S}_h$ and $|\{x \in \mathcal{D}_{h,s} : \tilde{f}_h(x) = s'\}| > 3/5|\mathcal{D}_{h,s}|$ **then**

4:          Swap the output of $s'$ with $s$ in $\tilde{f}_h$;

5:      **end if**

6: **end for**

7: **Output:** $\tilde{f}_{[H+1]}$

---

until $\mathscr{A}$ halts. Note that this procedure is essentially what practitioners use in Tang et al. (2017); Bellemare et al. (2016) as we have discussed in the beginning.

We now describe our algorithm in more detail. Suppose the algorithm $\mathscr{A}$ runs for $K$ episodes. At the beginning of each episode $k \in [K]$, $\mathscr{A}$ proposes a policy $\pi^k : [H] \times \mathcal{S} \to \mathcal{A}$ for the underlying MDP. Then we use the Trajectory Sampling Routine (TSR) to generate a trajectory $\tau^k$ using $\pi^k$ and then feed $\tau^k$ to $\mathscr{A}$. After $K$ episodes, we obtain a policy $\pi^{K+1}$ from $\mathscr{A}$ and a set of decoding functions $f^{K+1}_{[H+1]}$ from TSR. We then construct a policy for the BMDP as $\pi^{K+1} \circ f^{K+1}_{[H+1]}$. We repeat this process for $N$ times for making sure our algorithm succeeds with high probability.

The detailed description of TSR is displayed in Algorithm 2. We here briefly explain the idea. To distinguish between episodes, with input policy $\pi^k$ (Line 6 Algorithm 1), we add the episode index $k$ as superscripts to $\pi$ and $f_{[H+1]}$ in TSR. We maintain a policy set in memory and initialize it as an empty set at the beginning of Algorithm 1. Note that, at each episode, our goal is to simulate a trajectory of $\pi$ running on the underlying MDP. TSR achieves this in an iterative fashion: it starts with

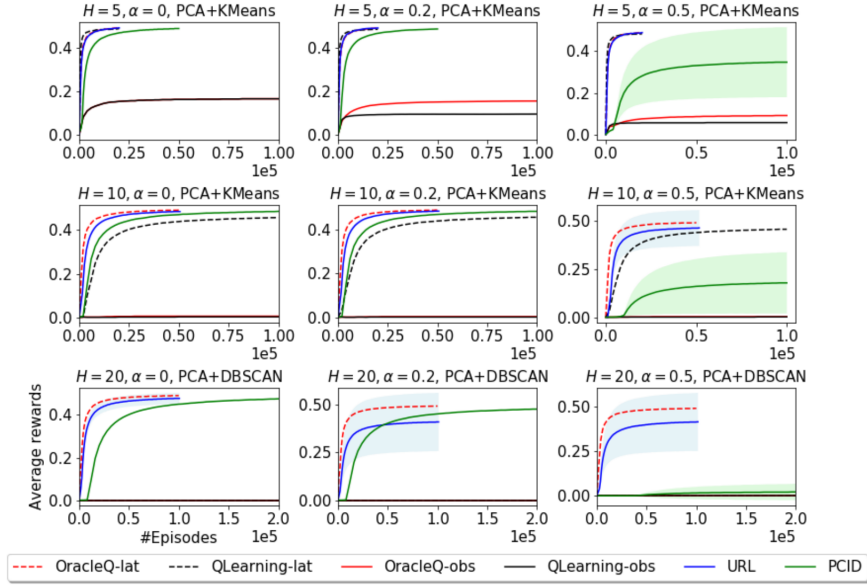

Figure 1: Performances for LockBernoulli. All lines are mean values of 50 tests and the shaded areas depict the one standard deviations.

the input policy $\pi^k$ and the latest-learned decoding functions $f_{[H+1]}^{k,0} := f_{[H+1]}^{k-1,J}$; for each iteration $i$, it first adds the policy $\pi^k \circ f_{[H+1]}^{k,i-1}$ in $\Pi$ and then plays $\mathrm{Unif}(\Pi)$ to collect a set of observation trajectories (i.e., each trajectory is generated by first uniformly randomly selecting a policy from $\Pi$ and then running it in the BMDP);[3] then updates $f_{[H+1]}^{k,i-1}$ to $\tilde{f}_{[H+1]}^{k,i}$ by running $\mathcal{ULO}$ on these collected observations. Note that $\mathcal{ULO}$ may output labels inconsistent with previously trained decoding functions. We further match labels of $\tilde{f}_{[H+1]}^{k,i}$ with the former ones by calling the FixLabel routine (Algorithm 3). To accomplish the label matching process, we cache a set $\mathcal{Z}$ in memory which stores observation examples $\mathcal{D}_{h,s}$ for each state $s$ and each level $h$. $\mathcal{Z}$ is initialized as an empty set and gradually grows. Whenever we confirm a new label, we add the corresponding observation examples to $\mathcal{Z}$ (Line 11 Algorithm 2). Then for later learned decoding functions, they can use this standard set to correspondingly swap their labels and match with previous functions. After the matching step, we get $f_{[H+1]}^{k,i}$. Continuously running for $J$ iterations, we stop and use $\pi^k \circ f_{[H+1]}^{k,J}$ to obtain a trajectory.

We now present our main theoretical result.

**Theorem 1.** *Suppose in Definition 1, $g^{-1}(\epsilon, \delta_1) = \mathrm{poly}(|\mathcal{S}|, 1/\epsilon, \log(\delta_1^{-1}))$ for any $\epsilon, \delta_1 \in (0,1)$ and $\mathscr{A}$ is $(\varepsilon, \delta_2)$-correct with sample complexity $\mathrm{poly}\left(|\mathcal{S}|, |\mathcal{A}|, H, 1/\varepsilon, \log\left(\delta_2^{-1}\right)\right)$ for any $\varepsilon, \delta_2 \in (0,1)$. Then Algorithm 1 outputs a policy $\phi$ such that with probability at least $1 - \delta$, $\phi$ is an $\varepsilon$-optimal policy for the BMDP, using at most $\mathrm{poly}\left(|\mathcal{S}|, |\mathcal{A}|, H, 1/\varepsilon, \log(\delta^{-1})\right)$ trajectories.*

We defer the proof to Appendix A. Theorem 1 formally justifies what we claimed in Section 1 that as long as the sample complexity of $\mathcal{ULO}$ is polynomial and $\mathscr{A}$ is a no-regret tabular RL algorithm, polynomial number of trajectories suffices to find a near-optimal policy. To our knowledge, this is the first result that proves unsupervised learning can guide exploration in RL problems with a huge observation space.

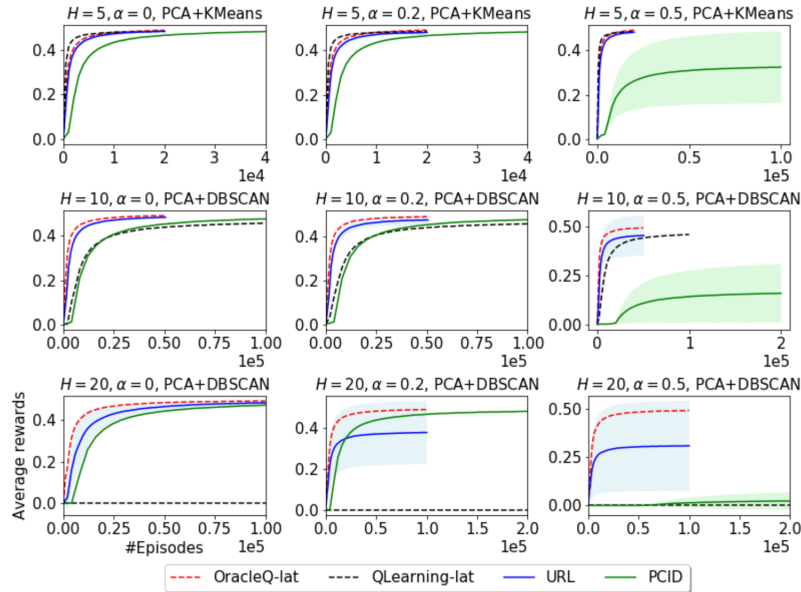

Figure 2: Performances for LockGaussian, $\sigma = 0.1$. All lines are mean values of $50$ tests and the shaded areas depict the one standard deviations. OracleQ-lat and QLearning-lat have direct access to the latent states, which are not for practical use. URL and PCID only have access to the observations. OracleQ-obs and QLearning-obs are omitted due to infinitely many observations.

## 5    Numerical Experiments

In this section we conduct experiments to demonstrate the effectiveness of our framework. Our code is available at `https://github.com/FlorenceFeng/StateDecoding`.

**Environments**    We conduct experiments in two environments: LockBernoulli and LockGaussian. These environments are also studied in Du et al. (2019a), which are designed to be hard for exploration. Both environments have the same latent state structure with $H$ levels, 3 states per level and 4 actions. At level $h$, from states $s_{1,h}$ and $s_{2,h}$ one action leads with probability $1 - \alpha$ to $s_{1,h+1}$ and with probability $\alpha$ to $s_{2,h+1}$, another has the flipped behavior, and the remaining two lead to $s_{3,h+1}$. All actions from $s_{3,h}$ lead to $s_{3,h+1}$. Non-zero reward is only achievable if the agent can reach $s_{1,H+1}$ or $s_{2,H+1}$ and the reward follows Bernoulli(0.5). Action labels are randomly assigned at the beginning of each time of training. We consider three values of $\alpha$: 0, 0.2, and 0.5.

In LockBernoulli, the observation space is $\{0,1\}^{H+3}$ where the first 3 coordinates are reserved for the one-hot encoding of the latent state and the last $H$ coordinates are drawn i.i.d from Bernoulli(0.5). LockBernoulli meets our requirements as a BMDP. In LockGaussian, the observation space is $\mathbb{R}^{H+3}$. Every observation is constructed by first letting the first three coordinates be the one-hot encoding of the latent state, then adding i.i.d Gaussian noises $\mathcal{N}(0, \sigma^2)$ to all $H + 3$ coordinates. We consider $\sigma = 0.1$ and $0.2$. LockGaussian is not a BMDP. We use this environment to evaluate the robustness of our method to violated assumptions.

The environments are designed to be hard for exploration. There are in total $4^H$ choices of actions of one episode, but only $2^H$ of them lead to non-zero reward in the end. So random exploration requires exponentially many trajectories. Also, with a larger $H$, the difficulty of learning accurate decoding functions increases and makes exploration with observations a more challenging task.

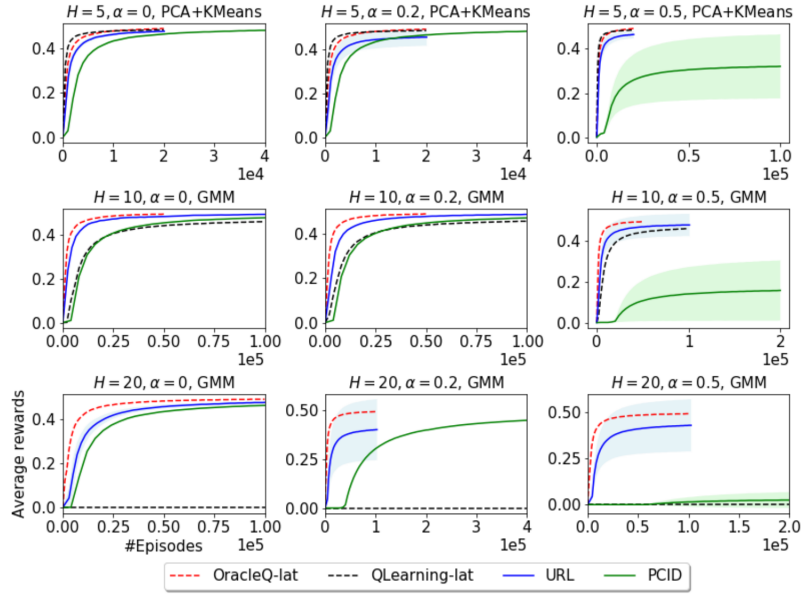

Figure 3: Performances for LockGaussian, $\sigma = 0.2$. All lines are mean values of 50 tests and the shaded areas depict the one standard deviations. OracleQ-lat and QLearning-lat have direct access to the latent states, which are not for practical use. URL and PCID only have access to the observations. OracleQ-obs and QLearning-obs are omitted due to infinitely many observations.

**Algorithms and Hyperparameters** We compare 4 algorithms: OracleQ (Jin et al., 2018); QLearning, the tabular Q-Learning with $\epsilon$-greedy exploration; URL, our method; and PCID (Du et al., 2019a). For OracleQ and QLearning, there are two implementations: 1. they directly see the latent states (OracleQ-lat and QLearning-lat); 2. only see observations (OracleQ-obs and QLearning-obs). For URL and PCID, only observations are available. OracleQ-lat and QLearning-lat are served as a near-optimal skyline and a sanity-check baseline to measure the efficiency of observation-only algorithms. OracleQ-obs and QLearning-obs are only tested in LockBernoulli since there are infinitely many observations in LockGaussian. For URL, we use OracleQ as the tabular RL algorithm. Details about hyperparameters and unsupervised learning oracles in URL can be found in Appendix C.

**Results** The results are presented in Figure 1, 2, and 3. $x$-axis is the number of training trajectories and $y$-axis is average reward. All lines are mean values of 50 tests and the shaded areas depict the one standard deviations. The title for each subfigure records problem parameters and the unsupervised learning method we apply for URL. In LockBernoulli, OracleQ-obs and QLearning-obs are far from being optimal even for small-horizon cases. URL is mostly as good as the skyline (OracleQ-lat) and much better than the baseline (QLearning-lat) especially when $H = 20$. URL outperforms PCID in most cases. When $H = 20$, we observe a probability of $80\%$ that URL returns near-optimal values for $\alpha = 0.2$ and $0.5$. In LockGaussian, for $H = 20$, we observe a probability of $> 75\%$ that URL returns a near-optimal policy for $\alpha = 0.2$ and $0.5$.

# 6 Conclusion

The current paper gave a general framework that turns an unsupervised learning algorithm and a no-regret tabular RL algorithm into an algorithm for RL problems with huge observation spaces. We provided theoretical analysis to show it is provably efficient. We also conducted numerical experiments to show the effectiveness of our framework in practice. An interesting future theoretical direction is to characterize the optimal sample complexity under our assumptions.

## Broader Impact

Our research broadens our understanding on the use of unsupervised learning for RL, which in turn can help researchers design new principled algorithms and incorporate safety considerations for more complicated problems.

We do not believe that the results in this work will cause any ethical issue, or put anyone at a disadvantage in the society.

## Acknowledgments and Disclosure of Funding

Fei Feng was supported by AFOSR MURI FA9550-18-10502 and ONR N0001417121. This work was done while Simon S. Du was at the Institute for Advanced Study and he was supported by NSF grant DMS-1638352 and the Infosys Membership.

## Footnotes

[2]For the convenience of analysis, we learn decoding functions for each level separately. In practice, observation data can be mixed up among all levels and used to train one decoding function for all levels.

[3]This resampling over all previous policies is mainly for the convenience of analysis. It can be replaced using previous data but requires more refined analysis.

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
