[Supplementary Material]

# A   Proofs for the Main Result

We first give a sketch of the proof. Note that if TSR always correctly simulates a trajectory of $\pi^k$ on the underlying MDP, then by the correctness of $\mathscr{A}$, the output policy of $\mathscr{A}$ in the end is near-optimal with high probability. If in TSR, $f_{[H+1]}^{k,J}$ decodes states correctly (up to a fixed permutation, with high probability) for every observation generated by playing $\pi^k \circ f_{[H+1]}^{k,J}$, then the obtained trajectory (on $\mathcal{S}$) is as if obtained with $\pi^k \circ f_{[H+1]}$ which is essentially equal to playing $\pi^k$ on the underlying MDP. Let us now consider $\pi^k \circ f_{[H+1]}^{k,i}$ for some intermediate iteration $i \in [J]$. If there are many observations from a previously unseen state, $s$, then $\mathcal{ULO}$ guarantees that all the decoding functions in future iterations will be correct with high probability of identifying observations of $s$. Since there are at most $|\mathcal{S}|$ states to reach for each level following $\pi^k$, after $(H+1)|\mathcal{S}|$ iterations, TSR is guaranteed to output a set of decoding functions that are with high probability correct under policy $\pi^k$. With this set of decoding functions, we can simulate a trajectory for $\mathscr{A}$ as if we know the true latent states.

For episode $k$, we denote the training dataset $\mathcal{D}$ generated by running $\mathrm{Unif}(\Pi)$ as $\{\mathcal{D}_{k,i,h}\}_{h=1}^{H+1}$ (Line 5) and the testing dataset $\mathcal{D}''$ generated by $\pi^k \circ f_{[H+1]}^{k,i-1}$ as $\{\mathcal{D}''_{k,i,h}\}_{h=1}^{H+1}$ (Line 6). The subscript $h$ represents the level of the observations. Furthermore, we denote by $\mu_{k,i,h}(\cdot)$ the distribution over hidden states at level $h$ induced by $\pi^k \circ f_{[H+1]}^{k,i-1}$. To formally prove the correctness of our framework, we first present the following lemma, showing that whenever some policy $\pi$ with some decoding functions visits a state $s$ with relatively high probability, all the decoding functions of later iterations will correctly decode the observations from $s$ with high probability.

**Lemma 1.** *Suppose for some $s^* \in \mathcal{S}_h$, $(k,i)$ is the earliest pair such that $\left|\{x \in \mathcal{D}''_{k,i,h} : f_h^{k,i}(x) = \alpha_h(s^*)\}\right| \geq 3\epsilon \cdot B \log(\delta_1^{-1})$ and $\{x \in \mathcal{D}''_{k,i,h} : f_h^{k,i}(x) = \alpha_h(s^*)\}$ is added into $\mathcal{Z}_h$ as $\mathcal{D}_{h,\alpha_h(s^*)}$ at line 11 Algorithm 2, where $\alpha_h$ is a good permutation between $f_h^{k,i}$ and $f_h$. Then for each $(k',i') > (k,i)$ (in lexical order), with probability at least $1 - \mathcal{O}(\delta_1)$,*

$$\Pr_{x \sim q(\cdot|s^*)} \left[f_h^{k',i'}(x) \neq \alpha_h^*(s^*)\right] \leq \epsilon$$

*provided $0 < \epsilon \log(\delta_1^{-1}) \leq 0.1$ and $B \geq B_0$. Here $B_0$ is some constant to be determined later and $\alpha_h^*$ is some fixed permutation on $\mathcal{S}_h$.*

*Proof of Lemma 1.* For iterations $(k',i') \geq (k,i)$, the function $\tilde{f}_h^{k',i'}$ is obtained by applying $\mathcal{ULO}$ on the dataset generated by

$$\mu' := \mathrm{Unif}(\{\mu_{k'',i'',h}\}_{(k'',i'')<(k',i')})$$

and the dataset has size $\left((k'-1) \cdot J + i'\right) \cdot B = \Theta(k'JB)$. Thus, with probability at least $1 - \delta_1$, for some permutation $\alpha_h'$,

$$\Pr_{s \sim \mu', x \sim q(\cdot|s)} \left[\tilde{f}_h^{k',i'}(x) \neq \alpha_h' \circ f_h(x)\right] \leq g\left(\Theta(k'JB), \delta_1\right). \tag{1}$$

By taking

$$B_0 := \Theta\left(\frac{g^{-1}(\epsilon^2/(K \cdot J), \delta_1)}{K \cdot J}\right), \tag{2}$$

we have when $B \geq B_0$, $g\left(\Theta(k'JB), \delta_1\right) \leq \epsilon^2/(K \cdot J)$ for all $k' \in [K]$. Later, in Proposition 1, we will show that $B_0 = \mathrm{poly}(|\mathcal{S}|, |\mathcal{A}|, H, 1/\varepsilon)$. Now we consider $f_h^{k,i}$. Since the FixLabel routine (Algorithm 3) does not change the accuracy ratio, from Equation (1), it holds with probability at least $1 - \delta_1$ that

$$\Pr_{s \sim \mu_{k,i,h}, x \sim q(\cdot|s)}[f_h^{k,i}(x) \neq \alpha_h \circ f_h(x)] \leq k \cdot J \cdot g\left(\Theta(kJB), \delta_1\right) \leq \epsilon.$$

Therefore, by Chernoff bound, with probability at least $1 - \mathcal{O}(\delta_1)$,

$$\left|\{x \in \mathcal{D}''_{k,i,h} : f_h(x) \neq s \text{ and } f_h^{k,i}(x) = \alpha_h(s)\}\right| < \epsilon \cdot B \log(\delta_1^{-1}).$$

Since $\left|\{x \in \mathcal{D}''_{k,i,h} : f^{k,i}_h(x) = \alpha_h(s^*)\}\right| \geq 3\epsilon \cdot B \log(\delta_1^{-1})$, we have that

$$\left|\{x \in \mathcal{D}''_{k,i,h} : \; f_h(x) = s^* \text{ and } f^{k,i}_h(x) = \alpha_h(s^*)\}\right| > \frac{2}{3} \cdot \left|\{x \in \mathcal{D}''_{k,i,h} : f^{k,i}_h(x) = \alpha_h(s^*)\}\right| \quad (3)$$
$$\geq 2\epsilon \cdot B \log(\delta_1^{-1}).$$

Thus, by Chernoff bound, with probability at least $1 - \mathcal{O}(\delta_1)$, $\mu_{k,i,h}(s^*) \geq \epsilon \cdot \log(\delta_1^{-1})$. Also note that $f^{k,i}_h$ is the first function that has confirmed on $s^*$ (i.e., no $\mathcal{D}_{h,\alpha_h(s^*)}$ exists in $\mathcal{Z}_h$ of line 8 at iteration $(k,i)$). By Line 10 and Line 11, for later iterations, in $\mathcal{Z}_h$, $\mathcal{D}_{h,\alpha_h(s^*)} = \{x \in \mathcal{D}''_{k,i,h} : f^{k,i}_h(x) = \alpha_h(s^*)\}$.

Next, for another $(k', i') > (k, i)$, we let the corresponding permutation be $\alpha'_h$ for $\tilde{f}^{k',i'}_h$. Since $\mu'(s') \geq \mu_{k,i,h}(s')/(k' \cdot J)$, with probability at least $1 - \delta_1$,

$$\Pr_{s \sim \mu_{k,i,h}, x \sim q(\cdot|s)} \left[\tilde{f}^{k',i'}_h(x) \neq \alpha'_h \circ f_h(x)\right] \leq k' \cdot J \cdot g(\Theta(k'JB), \delta_1).$$

Notice that

$$\Pr_{s \sim \mu_{k,i,h}, x \sim q(\cdot|s)} \left[\tilde{f}^{k',i'}_h(x) \neq \alpha'_h \circ f_h(x)\right] = \sum_{s' \in \mathcal{S}_h} \mu_{k,i,h}(s') \Pr_{x \sim q(\cdot|s')} \left[\tilde{f}^{k',i'}_h(x) \neq \alpha'_h \circ f_h(x)\right]$$
$$\geq \mu_{k,i,h}(s^*) \Pr_{x \sim q(\cdot|s^*)} \left[\tilde{f}^{k',i'}_h(x) \neq \alpha'_h \circ f_h(x)\right]$$
$$\geq \epsilon \cdot \log(\delta^{-1}) \Pr_{x \sim q(\cdot|s^*)} \left[\tilde{f}^{k',i'}_h(x) \neq \alpha'_h \circ f_h(x)\right].$$

Thus, with probability at least $1 - \delta_1$,

$$\Pr_{x \sim q(\cdot|s^*)} \left[\tilde{f}^{k',i'}_h(x) \neq \alpha'_h \circ f_h(x)\right] \leq \frac{k' \cdot J \cdot g(\Theta(k'JB), \delta_1)}{\epsilon \cdot \log(\delta_1^{-1})} \leq \epsilon$$

with $B \geq B_0$ and $B_0$ as defined in Equation (2). Let $s' := \alpha'_h(s^*)$. Conditioning on $\mathcal{ULO}$ being correct on $\tilde{f}^{k',i'}_{[H+1]}$ and $f^{k,i}_{[H+1]}$, by Chernoff bound and Equation (3), with probability at least $1 - \mathcal{O}(\delta_1)$, we have

$$\left|\{x \in \mathcal{D}_{h,\alpha_h(s^*)} : \tilde{f}^{k',i'}_h(x) = s'\}\right| \geq \left|\{x \in \mathcal{D}_{h,\alpha_h(s^*)} : f_h(x) = s^*, \tilde{f}^{k',i'}_h(x) = s'\}\right|$$
$$\geq (1 - \epsilon \cdot \log(\delta_1^{-1})) \cdot \frac{2}{3} \cdot \left|\mathcal{D}_{h,\alpha_h(s^*)}\right| > \frac{3}{5}\left|\mathcal{D}_{h,\alpha_h(s^*)}\right|,$$

where the fraction $\frac{2}{3}$ follows from Equation (3) and we use the fact that $\mathcal{D}''_{k,i,h}$ are independent from the training dataset. By our label fixing procedure, we find a permutation that swaps $s'$ with $s$ for $\tilde{f}^{k',i'}_h$ to obtain $f^{k',i'}_h$. By the above analysis, with probability at least $1 - \mathcal{O}(\delta_1)$, $\Pr_{x \sim q(\cdot|s^*)} \left[f^{k',i'}_h(x) \neq \alpha_h(s^*)\right] \leq \epsilon$ as desired. Consequently, we let $\alpha^*_h(s^*) = \alpha_h(s^*)$, which satisfies the requirement of the lemma. ∎

Next, by the definition of our procedure of updating the label standard dataset (Line 11, Algorithm 2), we have the following corollary.

**Corollary 1.** *Consider Algorithm 2. Let $\mathcal{Z}_{k,i,h}$ be the label standard dataset at episode $k$ before iteration $i$ for $\mathcal{S}_h$. Then, with probability at least $1 - \mathcal{O}(H|\mathcal{S}|\delta_1)$,*

$$\text{for all } k, i \text{ and } \mathcal{D}_{h,s} \in \mathcal{Z}_{k,i,h}, |\{x \in \mathcal{D}_{h,s} : \alpha^*_h \circ f_h(x) = s, s \in \mathcal{S}_h\}| > 2/3|\mathcal{D}_{h,s}|.$$

At episode $k$ and iteration $i$ of the algorithm TSR, let $\mathcal{E}_{k,i}$ be the event that for all $h \in [H+1], \mathcal{D}_{h,s} \in \mathcal{Z}_{k,i,h}, \Pr_{x \sim q(\cdot|s)} \left[f^{k,i}_h(x) \neq \alpha^*_h \circ f_h(x)\right] \leq \epsilon$. We have the following corollary as a consequence of Lemma 1 by taking the union bound over all states.

**Corollary 2.** $\forall k, i : \quad \Pr\left[\mathcal{E}_{k,i}\right] \geq 1 - \mathcal{O}(H|\mathcal{S}|\delta_1).$

The next lemma shows that after $(H+1)|\mathcal{S}| + 1$ iterations of the TSR subroutine, the algorithm outputs a trajectory for the algorithm $\mathscr{A}$ as if it knows the true mapping $f_{[H+1]}$.

**Lemma 2.** *Suppose in an episode $k$, we are running algorithm* TSR. *Then after $J = (H+1)|\mathcal{S}| + 1$ iterations, we have, for every $j \geq J$, with probability at least $1 - \mathcal{O}(H|\mathcal{S}|\delta_1)$,*

$$\textit{for all } h \in [H+1], \quad \Pr_{s \sim \mu_{k,j+1,h}, x \sim q(\cdot|s)} \left[ f_h^{k,j}(x) \neq \alpha_h^* \circ f_h(x) \right] \leq \epsilon'$$

*for some small enough $\epsilon$ and $50H \cdot \epsilon \cdot |\mathcal{S}| \cdot \log(\delta_1^{-1}) < \epsilon' < 1/2$, provided $B \geq B_0$ as defined in Lemma 1.*

*Proof of Lemma 2.* For $i < J$, there are two cases:

1. there exists an $h \in [H+1]$ such that $\Pr_{s \sim \mu_{k,i+1,h}, x \sim q(\cdot|s)} \left[ f_h^{k,i}(x) \neq \alpha_h \circ f_h(x) \right] > \epsilon'/(2H)$;

2. for all $h \in [H+1]$, $\Pr_{s \sim \mu_{k,i+1,h}, x \sim q(\cdot|s)} \left[ f_h^{k,i}(x) \neq \alpha_h \circ f_h(x) \right] \leq \epsilon'/(2H)$,

where $\alpha_h$ is some good permutations between $f_h^{k,i}$ and $f_h$. If case 1 happens, then there exists a state $s^* \in \mathcal{S}_h$ such that

$$\Pr_{x \sim q(\cdot|s^*)} \left[ f_h^{k,i}(x) \neq \alpha_h \circ f_h(x) \right] \cdot \mu_{k,i+1,h}(s^*) > \frac{\epsilon'}{2H|\mathcal{S}|}. \tag{4}$$

If $\mathcal{D}_{h,\alpha_h(s^*)} \in \mathcal{Z}_{k,i,h}$, where $\mathcal{Z}_{k,i,h}$ is defined as in Corollary 1, by Lemma 1, with probability at least $1 - \mathcal{O}(\delta_1)$,

$$\Pr_{x \sim q(\cdot|s^*)} [f_h^{k,i}(x) \neq \alpha_h^* \circ f_h(x)] \leq \epsilon$$

and $\alpha_h^*(s^*) = \alpha_h(s^*)$. Thus, $\mu_{k,i+1,h}(s^*) > \frac{\epsilon'}{2H|\mathcal{S}|}/\epsilon > 1$, a contradiction with $\mu_{k,i+1,h}(s^*) \leq 1$. Therefore, there is no $\mathcal{D}_{h,\alpha_h(s^*)}$ in $\mathcal{Z}_{k,i,h}$. Then, due to $\Pr_{x \sim q(\cdot|s^*)} \left[ f_h^{k,i}(x) \neq \alpha_h \circ f_h(x) \right] \leq 1$, by Equation (4), we have

$$\mu_{k,i+1,h}(s^*) > \frac{\epsilon'}{2H|\mathcal{S}|}. \tag{5}$$

Since $f_h^{k,i+1}$ is trained on $\mathrm{Unif}(\{\mu_{k',i',h}\}_{(k',i')<(k,i+1)})$, by Definition of $\mathcal{ULO}$, with probability at least $1 - \delta_1$,

$$\Pr_{s \sim \mu_{k,i+1,h}, x \sim q(\cdot|s)} \left[ f_h^{k,i+1}(x) \neq \alpha_h'(s) \right] \leq k \cdot J \cdot g(\Theta(kJB), \delta_1) \leq \epsilon^2,$$

with $B \geq B_0$ ($B_0$ is defined in Equation (2)) and $\alpha_h'$ is some good permutation between $f_h^{k,i+1}$ and $f_h$. Thus, by Equation (5) and the choice of $\epsilon$ and $\epsilon'$, we have

$$\Pr_{x \sim q(\cdot|s^*)} \left[ f_h^{k,i+1}(x) \neq \alpha_h'(s^*) \right] < \epsilon/25.$$

Thus,

$$\mu_{k,i+1,h}(s^*) \cdot \Pr_{x \sim q(\cdot|s^*)} \left[ f_h^{k,i+1}(x) = \alpha_h'(s^*) \right] > \frac{\epsilon'}{2H|\mathcal{S}|} \cdot (1 - \epsilon/25) > 24\epsilon \cdot \log(\delta_1^{-1}),$$

where the last inequality is due to $\epsilon < \epsilon' < 1$. By Chernoff bound, with probability at least $1 - \mathcal{O}(\delta_1)$,

$$|\{x \in \mathcal{D}_{k,i+1,h}'' : f_h^{k,i+1}(x) = \alpha_h'(s^*)\}| \geq 3\epsilon \cdot B \log(\delta_1^{-1}).$$

Therefore, if case 1 happens, one state $s$ will be confirmed in iteration $i+1$ and $\alpha_h^*(s^*) = \alpha_h'(s^*)$ is defined.

To analyze case 2, we first define sets $\{\mathcal{G}_{k,i+1,h}\}_{h=1}^{H+1}$ with $\mathcal{G}_{k,i+1,h} := \{s \in \mathcal{S}_h \mid \mathcal{D}_{h,s} \in \mathcal{Z}_{k,i+1,h}\}$, i.e., $\mathcal{G}_{k,i+1,h}$ contains all confirmed states of level $h$ before iteration $i+1$ at episode $k$. If case 2 happens, we further divide the situation into two subcases:

a) for all $h \in [H+1]$, for all $s \in \mathcal{G}_{k,i+1,h}^c$, $\mu_{k,i+1,h}(s) \leq \epsilon'/(8H|\mathcal{S}|)$;

b) there exists an $h \in [H+1]$ and a state $s^* \in \mathcal{G}^c_{k,i+1,h}$ such that $\mu_{k,i+1,h}(s^*) \geq \epsilon'/(8H|\mathcal{S}|)$,

First notice that for every $h \in [H+1]$ and $j > i$, since $f_h^{k,j}$ is trained on $\mathrm{Unif}(\{\mu_{k',i',h}\}_{(k',i')\leq(k,j)})$, by Definition of $\mathcal{ULO}$ and our choice of $B$ in Equation (2), with probability at least $1 - \delta_1$, we have

$$\Pr_{s\sim\mu_{k,i+1,h},x\sim q(\cdot|s)}[f_h^{k,j} \neq \alpha'_h(s)] \leq \epsilon^2, \tag{6}$$

$$\Rightarrow \sum_{s\in\mathcal{G}_{k,i+1,h}} \mu_{k,i+1,h}(s) \Pr_{x\sim q(\cdot|s)}[f_h^{k,j}(x) \neq \alpha'_h(s)] + \sum_{s\notin\mathcal{G}_{k,i+1,h}} \mu_{k,i+1,h}(s) \Pr_{x\sim q(\cdot|s)}[f_h^{k,j}(x) \neq \alpha'_h(s)] \leq \epsilon^2,$$

where $\alpha'_h$ is some good permutation between $f_h^{k,j}$ and $f_h$.

If subcase a) happens, note that for $s \in \mathcal{G}_{k,i+1,h}$, due to the FixLabel routine (Algorithm 3), $\alpha'_h(s) = \alpha^*_h(s)$, for $f_h^{k,j} (j > i)$ we have

$$\sum_{s\in\mathcal{S}_h} \mu_{k,i+1,h}(s) \Pr_{x\sim q(\cdot|s)}[f_h^{k,j}(x) \neq \alpha^*_h(s)]$$

$$= \sum_{s\in\mathcal{G}_{k,i+1,h}} \mu_{k,i+1,h}(s) \Pr_{x\sim q(\cdot|s)}[f_h^{k,j}(x) \neq \alpha^*_h(s)] + \sum_{s\notin\mathcal{G}_{k,i+1,h}} \mu_{k,i+1,h}(s) \Pr_{x\sim q(\cdot|s)}[f_h^{k,j}(x) \neq \alpha^*_h(s)]$$

$$= \sum_{s\in\mathcal{G}_{k,i+1,h}} \mu_{k,i+1,h}(s) \Pr_{x\sim q(\cdot|s)}[f_h^{k,j}(x) \neq \alpha'_h(s)] + \sum_{s\notin\mathcal{G}_{k,i+1,h}} \mu_{k,i+1,h}(s) \Pr_{x\sim q(\cdot|s)}[f_h^{k,j}(x) \neq \alpha^*_h(s)]$$

$$\leq \epsilon^2 + \epsilon'/(8H) < \epsilon'/(4H).$$

Taking a union bound over all $f_{[H+1]}^{k,j}$, we have that for any $h \in [H+1]$, with probability at least $1 - \mathcal{O}(H\delta_1)$,

$$\Pr_{s\sim\mu_{k,j+1,h},x\sim q(\cdot|s)}[f_h^{k,j}(x) = \alpha^*_h(s)] \geq \Pr_{s\sim\mu_{k,j+1,h},x\sim q(\cdot|s)}[f_h^{k,j}(x) = \alpha^*_h(s) = f_h^{k,i}(x)]$$

$$\geq \Pr_{\text{for all } h'\in[h],s_{h'}\sim\mu_{k,j+1,h'},x_{h'}\sim q(\cdot|s_{h'})}[\text{ for all } h' \in [h], f_{h'}^{k,j}(x_{h'}) = \alpha^*_{h'}(s) = f_{h'}^{k,i}(x_{h'})]$$

$$= \Pr_{\text{for all } h'\in[h],s_{h'}\sim\mu_{k,i+1,h'},x_{h'}\sim q(\cdot|s_{h'})}[\text{ for all } h' \in [h], f_{h'}^{k,j}(x_{h'}) = \alpha^*_{h'}(s) = f_{h'}^{k,i}(x_{h'})]$$

$$\geq 1 - (\epsilon'/(2H) + \epsilon'/(4H)) \cdot H \geq 1 - \epsilon'.$$

Therefore, if case 2 and subcase a) happens, the desired result is obtained.

If subcase b) happens, we consider the function $f_h^{k,i+1}$. By Equation (6),

$$\mu_{k,i+1,h}(s^*) \cdot \Pr_{x\sim q(\cdot|s^*)}[f_h^{k,i+1}(x) \neq \alpha'_h(s^*)] \leq \epsilon^2$$

$$\Rightarrow \Pr_{x\sim q(\cdot|s^*)}[f_h^{k,i+1}(x) \neq \alpha'_h(s^*)] \leq \epsilon^2/(\epsilon'/(8H|\mathcal{S}|)) \leq \epsilon,$$

where $\alpha'_h$ here is some good permutation between $f_h^{k,i+1}$ and $f_h$. Thus,

$$\mu_{k,i+1,h}(s^*) \cdot \Pr_{x\sim q(\cdot|s^*)}[f_h^{k,i+1}(x) = \alpha'_h(s^*)] > \frac{\epsilon'}{8H|\mathcal{S}|} \cdot (1 - \epsilon) > 6\epsilon \cdot \log(\delta_1^{-1}).$$

By Chernoff bound, with probability at least $1 - \mathcal{O}(\delta_1)$, $|\{x \in \mathcal{D}''_{k,i+1,h} : f_h^{k,i+1}(x) = \alpha'_h(s^*)\}| \geq 3\epsilon \cdot B\log(\delta_1^{-1})$. Therefore, the state $s^*$ will be confirmed in iteration $i+1$ and $\alpha^*_h(s^*) = \alpha'_h(s^*)$ is defined.

In conclusion, for each iteration, there are two scenarios, either the desired result in Lemma 2 holds already or a new state will be confirmed for the next iteration. Since there are in total $\sum_{h=1}^{H+1} |\mathcal{S}_h| \leq (H+1)|\mathcal{S}|$ states, after $J := (H+1)|\mathcal{S}| + 1$ iterations, by Lemma 1, with probability at least

$1 - \mathcal{O}(H|\mathcal{S}|\delta_1)$, for every $j \geq J$, for all $h \in [H+1]$ and all $s \in \mathcal{S}_h$, we have $\Pr_{x \sim q(\cdot|s)}[f_h^{k,j}(x) \neq \alpha_h^*(s)] \leq \epsilon$. Therefore, it holds that for

$$\Pr_{s \sim \mu_{k,j+1,h}, x \sim q(\cdot|s)}(f_h^{k,j}(x) \neq \alpha_h^*(s)) \leq \epsilon < \epsilon'.$$

∎

**Proposition 1.** *Suppose in Definition 1, $g^{-1}(\epsilon, \delta_1) = \mathrm{poly}(1/\epsilon, \log(\delta_1^{-1}))$ for any $\epsilon, \delta_1 \in (0,1)$ and $\mathscr{A}$ is $(\varepsilon, \delta_2)$-correct with sample complexity $\mathrm{poly}\left(|\mathcal{S}|, |\mathcal{A}|, H, 1/\varepsilon, \log\left(\delta_2^{-1}\right)\right)$ for any $\varepsilon, \delta_2 \in (0,1)$. Then for each iteration of the outer loop of Algorithm 1, the policy $\phi^n$ is an $\varepsilon/3$-optimal policy for the BMDP with probability at least $0.99$, using at most $\mathrm{poly}\left(|\mathcal{S}|, |\mathcal{A}|, H, 1/\varepsilon\right)$ trajectories.*

*Proof of Proposition 1.* We first show that the trajectory obtained by running $\pi^k$ with the learned decoding functions $f_{[H+1]}^{k,J}$ matches, with high probability, that from running $\pi^k$ with $\alpha_{[H+1]}^* \circ f_{[H+1]}$. Let $K = C(\varepsilon/4, \delta_2)$ be the total number of episodes played by $\mathscr{A}$ to learn an $\varepsilon/4$-optimal policy with probability at least $1 - \delta_2$. For each episode $k \in [K]$, let the trajectory of observations be $\{x_h^k\}_{h=1}^{H+1}$. We define event

$$\mathcal{E}_k := \{\forall h \in [H+1], f_h^{k,J}(x_h^k) = \alpha_h^*(f_h(x_h^k))\},$$

where $J = (H+1)|\mathcal{S}| + 1$. Note that on $\mathcal{E}_k$, the trajectory of running $\pi^k \circ \alpha_{[H+1]}^* \circ f_{[H+1]}$ equals running $\pi^k \circ f_{[H+1]}^{k,J}$. We also let the event $\mathcal{F}$ be that $\mathcal{ULO}$ succeeds on every iteration (satisfies Lemma 2). Thus,

$$\Pr[\mathcal{F}] \geq 1 - K \cdot J \cdot \delta_1 = 1 - \mathrm{poly}(|\mathcal{S}|, |\mathcal{A}|, H, 1/\varepsilon, \log(\delta_1^{-1})) \cdot \delta_1.$$

Furthermore, each $x_{k,h}$ is obtained by the distribution $\sum_s \mu_{k,J+1,h}(s) q(\cdot|s)$. On $\mathcal{F}$, by Lemma 2, we have

$$\Pr[f_h^{k,J}(x_h^k) = \alpha_h^*(f_h(x_h^k))] \leq \epsilon'$$

by the choice of $B$. Therefore,

$$\Pr[\mathcal{E}_k|\mathcal{F}] \geq 1 - (H+1)\epsilon'.$$

Overall, we have

$$\Pr\left[\mathcal{E}_k, \ \forall k \in [K] \Big| \mathcal{F}\right] \geq 1 - K(H+1)\epsilon'.$$

Thus, with probability at least $1 - \delta_2 - \mathrm{poly}(|\mathcal{S}|, |\mathcal{A}|, H, 1/\varepsilon, \log(\delta_1^{-1})) \cdot (\epsilon' + \delta_1)$, $\mathcal{A}$ outputs a policy $\pi$, that is $\varepsilon/4$-optimal for the underlying MDP with state sets $\{\mathcal{S}_h\}_{h=1}^{H+1}$ permutated by $\alpha_{[H+1]}^*$, which we denote as event $\mathcal{E}'$. Conditioning on $\mathcal{E}'$, since on a high probability event $\mathcal{E}''$ with $\Pr[\mathcal{E}''] \geq 1 - (H+1)\epsilon'$, $\pi \circ f_{[H+1]}^{K,J}$ and $\pi \circ \alpha_{[H+1]}^* \circ f_{[H+1]}$ have the same trajectory, the value achieved by $\pi \circ f_{[H+1]}^{K,J}$ and $\pi \circ \alpha_{[H+1]}^* \circ f_{[H+1]}$ differ by at most $(H+1)^2 \epsilon'$. Thus, with probability at least $1 - \delta_2 - \mathrm{poly}(|\mathcal{S}|, |\mathcal{A}|, H, 1/\varepsilon, \log(\delta_1^{-1})) \cdot (\epsilon' + \delta_1)$, the output policy $\pi \circ f_{[H+1]}^{K,J}$ is at least $\varepsilon/4 + \mathcal{O}(H^2\epsilon')$ accurate, i.e.,

$$V_1^* - V_1^{\pi \circ f_{[H+1]}^{K,J}} \leq V_1^* - V_1^{\pi \circ \alpha_{[H+1]}^* \circ f_{[H+1]}} + \mathcal{O}(H^2\epsilon') \leq \varepsilon/4 + \mathcal{O}(H^2\epsilon').$$

Setting $\epsilon'$, $\delta_1$, and $\delta_2$ properly, $V_1^* - V_1^{\pi \circ f_{[H+1]}^{K,J}} \leq \varepsilon/3$ with probability at least $0.99$. Since $1/\delta_1 = \mathrm{poly}(|\mathcal{S}|, |\mathcal{A}|, H, 1/\varepsilon)$ and $1/\epsilon = \mathrm{poly}(|\mathcal{S}|, |\mathcal{A}|, H, 1/\varepsilon, \log(\delta_1^{-1}))$, $B_0$ in Lemma 1 and Lemma 2 is $\mathrm{poly}(|\mathcal{S}|, |\mathcal{A}|, H, 1/\varepsilon)$. The desired result is obtained. ∎

Finally, based on Proposition 1, we establish Theorem 1.

*Proof of Theorem 1.* By Proposition 1 and taking $N = \lceil \log(2/\delta)/2 \rceil$, with probability at least $1 - \delta/2$, there exists a policy in $\{\phi^n\}_{n=1}^N$ that is $\varepsilon/3$-optimal for the BMDP. For each policy $\phi^n$, we take $L := \lceil 9H^2/(2\varepsilon^2) \log(2N/\delta) \rceil$ episodes to evaluate its value. Then by Hoeffding's inequality, with probability at least $1 - \delta/(2N)$, $|\bar{V}_1^{\phi^n} - V_1^{\phi^n}| \leq \varepsilon/3$. By taking the union bound and selecting the policy $\phi \in \mathrm{argmax}_{\phi \in \phi[N]} \bar{V}_1^\phi$, with probability at least $1 - \delta$, it is $\varepsilon$-optimal for the BMDP. In total, the number of needed trajectories is $N \cdot \sum_{k=1}^K \sum_{i=1}^J \left((k-1)J + i + 1\right) B + N \cdot L = \mathcal{O}(N \cdot K^2 \cdot J^2 \cdot B + N \cdot L) = \mathrm{poly}(|\mathcal{S}|, |\mathcal{A}|, H, 1/\varepsilon, \log(\delta^{-1}))$. We complete the proof. ∎

# B Examples of Unsupervised Learning Oracle

In this section, we give some examples of $\mathcal{ULO}$. First notice that the generation process of $\mathcal{ULO}$ is termed as the mixture model in statistics (McLachlan and Basford, 1988; McLachlan and Peel, 2004), which has a wide range of applications (see e.g., Bouguila and Fan (2020)). We list examples of mixture models and some algorithms as candidates of $\mathcal{ULO}$.

**Gaussian Mixture Models (GMM)**   In GMM, $q(\cdot|s) = \mathcal{N}(s, \sigma_s^2)$, i.e., observations are hidden states plus Gaussian noise.[4] When the noises are (truncated) Gaussian, under certain conditions, e.g. states are well-separated, we are able to identify the latent states with high accuracy. A series of works (Arora and Kannan, 2001; Vempala and Wang, 2004; Achlioptas and McSherry, 2005; Dasgupta and Schulman, 2000; Regev and Vijayaraghavan, 2017) proposed algorithms that can be served as $\mathcal{ULO}$.

**Bernoulli Mixture Models (BMM)**   BMM is considered in binary image processing (Juan and Vidal, 2004) and texts classification (Juan and Vidal, 2002). In BMM, every observation is a point in $\{0, 1\}^d$. A true state determines a frequency vector. In Najafi et al. (2020), the authors proposed a reliable clustering algorithm for BMM data with polynomial sample complexity guarantee.

**Subspace Clustering**   In some applications, each state is a set of vectors and observations lie in the spanned subspace. Suppose for different states, the basis vectors differ under certain metric, then recovering the latent state is equivalent to subspace clustering. Subspace clustering has a variety of applications include face clustering, community clustering, and DNA sequence analysis (Wallace et al., 2015; Vidal, 2011; Elhamifar and Vidal, 2013). Proper algorithms for $\mathcal{ULO}$ can be found in e.g., (Wang et al., 2013; Soltanolkotabi et al., 2014).

In addition to the aforementioned models, other reasonable settings are Categorical Mixture Models (Bontemps and Toussile, 2013), Poisson Mixture Models (Li and Zha, 2006), Dirichlet Mixture Models (Dahl, 2006) and so on.

# C More about Experiments

**Parameter Tuning**   In the experiments, for OracleQ, we tune the learning rate and a confidence parameter; for QLearning, we tune the learning rate and the exploration parameter $\epsilon$; for PCID, we follow the code provided in Du et al. (2019a), tune the number of clusters for $k$-means and the number of trajectories $n$ to collect in each outer iteration, and finally select the better result between linear function and neural network implementation.

**Unsupervised Learning Algorithms**   In our method, we use OracleQ as the tabular RL algorithm to operate on the decoded state space and try three unsupervised learning approaches: 1. first conduct principle component analysis (PCA) on the observations and then use $k$-means (KMeans) to cluster; 2. first apply PCA, then use Density-Based Spatial Clustering of Applications with Noise (DBSCAN) for clustering, and finally use support vector machine to fit a classifier; 3. employ Gaussian Mixture Model (GMM) to fit the observation data then generate a label predictor. We call the python library `sklearn` for all these methods. During unsupervised learning, we do not separate observations by levels but add level information in decoded states. Besides the hyperparameters for OracleQ and the unsupervised learning oracle, we also tune the batch size $B$ adaptively in Algorithm 2. In our tests, instead of resampling over all previous policies as Line 5 Algorithm 2, we use previous data. Specifically, we maintain a training dataset $\mathcal{D}$ in memory and for iteration $i$, generate $B$ training trajectories following $\pi \circ f_{[H+1]}^{i-1}$ and merge them into $\mathcal{D}$ to train $\mathcal{ULO}$. Also, we stop training decoding functions once they become stable, which takes 100 training trajectories when $H = 5$, $500 \sim 1000$ trajectories when $H = 10$, and $1000 \sim 2500$ trajectories when $H = 20$. Since this process stops very quickly, we also skip the label matching steps (Line 8 to Line 12 Algorithm 2) and the final decoding function leads to a near-optimal performance as shown in the results.

## Footnotes

[4]To make the model satisfy the disjoint block assumption in the definition of BMDP, we need some truncation of the Gaussian noise so that each observation only corresponds to a unique hidden state.