[Reviews · NeurIPS 2020]

Review 1

Summary and Contributions: This paper introduces a method for efficient exploration in RL. The proposed method assumes an MDP with high-dimensional states that are generated by an underlying lower-dimensional process, such that these states can be compressed via an unsupervised learning algorithm/oracle. The method then (1) defines an MDP over the resulting low-dimensional state space; and (2) learns a policy by generating trajectories in low-dimensional space, which arguably facilitates exploration. At each iteration, the algorithm gathers data to compute a policy and also to improve the embedding model computed by the unsupervised algorithm. The authors show that as long as the unsupervised algorithm and the tabular RL algorithm have polynomial sample complexity, it is possible to find a near-optimal policy with polynomial complexity in the number of latent states, which is much smaller than the number of high-dimensional states. The authors show experiments on simple synthetic domains, where their framework presents good performance.

Strengths: This is a well-written (mostly theoretical) paper. Although it demonstrates empirically that the proposed method works in simple problems, the underlying theory seems sound and the formal results that are presented are not trivial.

Weaknesses: The theory underlying this method seems well-developed. I am curious, however, about whether it would perform well in more realistic problems. Your experiments are done in synthetic domains with highly structured high-dimensional states that, by construction, have a clear and well-defined low-dimensional representation. Most realistic problems where high-dimensional states are a problem, however, require the use of (non-linear) value function approximation, which (I believe) is not allowed in your framework. Do you believe that your method could nonetheless be applied to more realistic settings? Furthermore, it is not clear to me that the proposed method actually improves learning by (as the title suggests) exploring more efficiently. Is it fair/accurate to say that you are efficiently exploring? It seems like your method is not really guiding exploration, but it is instead solving a low-dimensional version of the original problem. Although this may accelerate learning, faster learning results from having of a smaller MDP, not from using a method that explicitly explores efficiently. In fact, the tabular algorithm could explore using a random policy and it would still have the desired properties required by your framework.

Correctness: The theory underlying the results seems correct. The empirical methodology seems correct.

Clarity: The paper is generally well-written.

Relation to Prior Work: Related work is properly discussed.

Reproducibility: Yes

Additional Feedback: This paper introduces a method for efficient exploration in RL. The proposed method assumes an MDP with high-dimensional states that are generated by an underlying lower-dimensional process, such that these states can be compressed via an unsupervised learning algorithm/oracle. The method then (1) defines an MDP over the resulting low-dimensional state space; and (2) learns a policy by generating trajectories in low-dimensional space, which arguably facilitates exploration. At each iteration, the algorithm gathers data to compute a policy and also to improve the embedding model computed by the unsupervised algorithm. The authors show that as long as the unsupervised algorithm and the tabular RL algorithm have polynomial sample complexity, it is possible to find a near-optimal policy with polynomial complexity in the number of latent states, which is much smaller than the number of high-dimensional states. The authors show experiments on simple synthetic domains, where their framework presents good performance. I have a few questions and comments: 1) What is the technical reason why the reward function needs to depend on the time step within an episode? 2) In most RL problems, the exact horizon H is unknown. Does your method support learning in those settings? 3) Are "levels" the same as time steps within the episode? 4) Related to the question above: my understanding is that you need to compute one embedding/unsupervised model per time step/level in the episode. If this is the case, why is this required? 5) On line 216, you say that the learning process needs to be executed for N times to ensure that the algorithm succeeds with high probability. Why do you need to run the algorithm several times, to avoid a failure, given that it should (in theory, if all of its components have polynomial sample complexity) always converge to a near-optimal solution? 6) Most unsupervised learning algorithms for computing embeddings return continuous low-dimensional representations of the input data. How do you address this issue, given that (if I understand this correctly) the policy for the underlying MDP M' needs to be identified by a tabular RL algorithm, which operates over discrete/finite state representations? 7) Is it fair/accurate to say that you are efficiently exploring? It seems like your method is not really guiding exploration, but it is instead solving a low-dimensional version of the original problem. Although this may accelerate learning, faster learning results from having of a smaller MDP, not from using a method that explicitly explores efficiently. In fact, the tabular algorithm could explore using a random policy and it would still have the desired properties required by your framework. 8) Your experiments are done in synthetic domains with highly structured high-dimensional states that, by construction, have a clear and well-defined low-dimensional representation. Most realistic problems where high-dimensional states are a problem, however, require the use of (non-linear) value function approximation, which (I believe) is not allowed in your framework. Do you believe that your method could nonetheless be applied to more realistic settings? ************************ POST-REBUTTAL COMMENTS I have read the authors' rebuttal and thank them for addressing my questions and suggestions. I believe that they have satisfactorily addressed the main points that I brought up, and I therefore keep my original overall positive view on this submission.


Review 2

Summary and Contributions: The paper studies exploration in reinforcement learning using unsupervised learning oracles. They derive a polynomial algorithm for Block Markov Decision Process when the unsupervised learning method has polynomial complexity. The method appears to be both theoretically as well as empirically strong. ======= Unchanged after rebuttal

Strengths: - the paper opens a new interesting research direction using unsupervised learning methods; this has not been studied before on the theoretical side for reinforcement learning exploration. - Removing the minimum visitation probably requirement of the prior work of Du et al '19 on block MDPs is another big contribution by itself.

Weaknesses: - no major limitations; this is a first step in a certain direction of research. The bound is vaguely stated as ``polynomial'' which means there is a lot of room for further studies in the area.

Correctness: Yes

Clarity: Yes

Relation to Prior Work: Yes

Reproducibility: Yes

Additional Feedback:


Review 3

Summary and Contributions: This work examines the utility of unsupervised learning for efficient exploration on reinforcement learning tasks. The proposed framework is based on two main components: an unsupervised learning algorithm and a no-regret tabular reinforcement learning algorithm. Roughly speaking, the unsupervised learning oracle encodes the state space in such a way that similar states are clustered together. Then, we are able to apply any no-regret tabular reinforcement learning algorithm on the new state space (it is referred as latent space in the paper). Authors prove that the proposed framework is able to find a near-optimal policy with sample complexity polynomial in the number of latent states, which is significantly smaller than the number of possible observations. Experiments have been conducted on two hard for exploration environments, the LockBernoulli and the LockGaussian, where different unsupervised learning algorithms and tabular RL algorithms have been examined.

Strengths: Authors propose a new algorithmic framework for the Block Markov Decision Process that is based on an unsupervised learning oracle and a tabular RL algorithm to find a near-optimal policy. The main contribution of this work is its theoretical analysis that proves the efficiency of the proposed algorithmic scheme. More specifically, it has been proved that the proposed framework is able to find a near-optimal policy with sample complexity polynomial in the number of latent states, which is significantly smaller than the number of possible observations. Moreover, the impact of the unsupervised algorithm on the performance of the proposed framework has been examined.

Weaknesses: The proposed framework assumes that we have access to an unsupervised learning oracle (ULO) and an (ε, δ)-correct episodic no-regret algorithm. In general this assumption is a little bit restricted as the performance of the proposed framework is highly depending on the ability of the unsupervised learning scheme to discover the latent space. Nevertheless, the theoretical analysis of the framework doesn't consider the probability the unsupervised learning oracle to encode inappropriately the observations. Also the training of the ULO is not explained in details. I think that the ULO is the key component of the proposed framework and it can have a huge impact on its performance. Moreover, authors should examine and/or to discuss its applicability of the proposed framework on more challenging environments.

Correctness: In my opinion the framework introduced in this work along with the presented empirical methodology are correct.

Clarity: In general the paper is well written and the reader can easily understand the main idea of the paper. The only part of the paper that need to be revised is the description of the label matching process. I found this part a little bit confused and some parts of Alg.2 should be explained in a more clear way. For instance, what is the purpose of the training data?

Relation to Prior Work: The related work is presented clearly in the paper and the author discuss in details the motivation and how this work differs from previous contributions.

Reproducibility: Yes

Additional Feedback: See my comments on the previous sections. I would like to thank authors for their rebuttal. Having also read other reviews, I decided to keep my original positive evaluation.


Review 4

Summary and Contributions: This paper proposes an algorithmic framework that can combine unsupervised learning oriacle with tabular RL method. There is theoretical guarantee that when both parts are in polynomial sample complexity, the near optimal policy can be found. The instantiation of this method ourperforms previous methods.

Strengths: This is the first provable method that can be applied in relatively large observation space. The unified framework is general can in theory can be used with any unsupervised learning methods and tabular RL algo.

Weaknesses: Some curves in the experiments are lack of error bar and clarification in the caption --- is this 1 standard deviation? More qualitative visualized demonstrations of the task would be helpful for understanding the environment.

Correctness: The proposed theory is correct from my perspective. And the experiments along with the theory confirm the claim of the authors. I believe the results are sound and valid.

Clarity: The paper is well written. The paper is written in formal and fluent language. There are small typos, however.

Relation to Prior Work: The discussion is clear and sufficient. The discussion includes previous works especially those that inspire this work. They further differentiate their method with previous method. I believe there are more that can be added such as the difference between their setting and real-world settings used in empirical papers. This can be really useful for practitioners to create new algorithms.

Reproducibility: Yes

Additional Feedback: Post rebuttal: Problem addressed. Recommend for acceptance.

[Author Response · NeurIPS 2020]

We want to thank all reviewers for their time and feedback.

**To Reviewer #1**:

* "Do you believe that your method could nonetheless be applied to more realistic settings?" **A1**: Yes. Our method is
inspired by the influential empirical paper by [Tang et al. 2017] as we mentioned in Line 188-190.

* "Is it fair/accurate to say that you are efficiently exploring?..." **A2**: Our method does guide exploration. Our
framework relies on a provably efficient tabular RL algorithm which has to have an efficient exploration component,
such as UCB or Thompson sampling. We want to emphasize that a tabular algorithm that uses random policy, e.g.,
$\epsilon$-greedy, *does not* have the desired properties we need because it requires an exponential number of samples. See the
lower bound in Section A of [Jin et al. 2018]. We will add more clarifications on this point.

* "Why the reward function depends on the time step?" **A3**: This setting is more general than the case where the
reward function is independent of the time step. This setting has also been studied in many previous works. We will add
more clarifications.

* "Does your method support learning if $H$ is unknown?" **A4**: Yes, our algorithm still works as long as we know an
upper bound of $H$. We will add discussions about this. Thanks for pointing it out.

* "Are levels the same as time steps within the episode?" **A5**: Yes.

* "...one embedding for each level?" **A6**: In our description of the framework, we need one decoding function per
level. But this is mainly for the convenience of analysis and is not necessary. Observation data can be mixed up among
levels and used to train ONE decoding function for all levels. This is what we implemented in the numerical test. We
will add more clarification about this part in the next version. Thank you for asking.

* "Why run $N$ times?" **A7**: If we directly require every output policy (Line 11, Algorithm 1) to be near-optimal with
probability $1 - \delta$, the sample complexity will be $\mathcal{O}(1/\delta)$ instead of $\mathcal{O}(\log(1/\delta))$. To avoid this, we only require each
output policy to be near-optimal with a parameter-free probability e.g., 0.99 and then we run $N = \mathcal{O}(\log(1/\delta))$ times
to select the best policy out of $N$ and achieve the final goal.

* "...continuous low-dimensional representations..." **A8**: We are using clustering methods on the low-dimensional
embedding space to reduce it to a finite-state space. This is also the implementation in [Tang et at. 2017].

**To Reviewer #3**: Thank you for the positive review.

**To Reviewer #4**:

* "...the framework doesn't consider the probability that the ULO encodes inappropriately the observations." **A1**: In
our definition of ULO (Definition 1), we do consider the probability that ULO encodes wrongly, which is described by
the term $g(n, \delta)$. Besides, in the definition, the correctness of a function $\hat{f}$ is on-average, i.e., if for some state $s$, $\mu(s)$ is
close to 0, $\hat{f}$ can be very inaccurate on $s$.

* "...the training of ULO is not explained in detail." **A2**: The training of ULO is described in Line 3-12 Algorithm
2. We also explained in Line 218-232. Due to the page limit, we defer some parts in the appendix. We use ULO as a
black-box oracle for abstraction and generality. We discuss some specific examples in Section B. We will add some
details to the main text in the final version. Thanks for asking.

* "...examine and/or discuss the applicability of the framework in more challenging environments." **A3**: We have more
discussion in the appendix about, for example, the choice of ULO one can consider solving real-world problems. We
will consider this in the next version.

* "...the label matching process and parts of Alg.2 should be explained in a more clear way. For instance, what is
the purpose of the training data?" **A4**: Thanks for the advice. Due to the page limit, we defer some descriptions to the
appendix. We will improve this part in the next version. For your question, the training data is collected to feed into the
ULO and generate a more refined decoding function for each iteration.

**To Reviewer #5**:

* "...some curves lack error bars and clarification in the caption." **A1**: Thanks for mentioning this. We double-checked
the graphs and we are sure that no curve lacks an error bar in the current version. They might be too tiny to observe.
Yes, it is 1 standard deviation. We will add more clarification in the caption in the next version.

* "There are some small typos." **A2**: Thank you. We will double-check and revise them in the next version.

* "More can be added such as the difference between their setting and real-world settings." **A3**: Very good advice. We
will consider this in the next version.

[Meta-Review · NeurIPS 2020]

The paper focuses on efficiently exploring MDPs with high dimensional state representations, by combining an unsupervised algorithm for learning a low-dimensional representation and then solving the problem in this low-dimensional space. The paper is largely theoretic and show that in certain conditions, near-optimal policies can be found with polynomial complexity in the number of latent states. The reviewers mostly agreed on the following points. The paper is considered well-written, and presents theoretically strong results that are sound, novel, and non-trivial. As weaknesses of the paper the reviewers mentioned the lack of empirical results in more realistic settings and restrictive assumptions. The reviewers had a couple of doubts about the paper, which mostly were addressed in the rebuttal (e.g. is improvement due to exploration, how is a discrete set of latent states, errorbars). Everything considered, in accordance with the opinion of the reviewer, I recommend acceptance for this paper. I would like to ask the authors to make sure the clarifications given in the rebuttal are reflected in the final version of the manuscript.